# Mediterranean Alcohol-Drinking Patterns and All-Cause Mortality in Women More Than 55 Years Old and Men More Than 50 Years Old in the “Seguimiento Universidad de Navarra” (SUN) Cohort

**DOI:** 10.3390/nu14245310

**Published:** 2022-12-14

**Authors:** María Barbería-Latasa, Maira Bes-Rastrollo, Rafael Pérez-Araluce, Miguel Ángel Martínez-González, Alfredo Gea

**Affiliations:** 1Department of Preventive Medicine and Public Health, University of Navarra, 31008 Pamplona, Spain; 2Instituto de Investigación Sanitaria de Navarra (IdiSNA), Navarra Institute for Health Research, 31008 Pamplona, Spain; 3Biomedical Research Network Center for Pathophysiology of Obesity and Nutrition (CIBEROBN), Carlos III Health Institute, 28029 Madrid, Spain

**Keywords:** MADP, Mediterranean diet, alcohol, drinking pattern, binge drinking, moderate drinking, elderly, mortality

## Abstract

Background: Most of the available epidemiological evidence on alcohol and chronic disease agrees on recommending alcohol abstention to young people, but some controversy exists about the most appropriate recommendation for alcohol abstention for people of older ages. A growing body of evidence suggests that the pattern of alcohol consumption is likely to be a strong effect modifier. The Mediterranean Alcohol Drinking Pattern (MADP) represents a score integrating several dimensions of drinking patterns (moderation, preference for red wine, drinking with meals, and avoiding binge drinking). Our aim was to clarify this issue and provide more precise recommendations on alcohol consumption. Methods: We prospectively followed-up 2226 participants (men older than 50 years and women older than 55 years at baseline) in the Seguimiento Universidad de Navarra (SUN) cohort. We classified participants into three categories of adherence to the MADP score (low, moderate, and high), and we added a fourth category for abstainers. Cox regression models estimated multivariable-adjusted hazard ratios (HR) of all-cause death and 95% confidence intervals (CI) using low MADP adherence as the reference category. Results: The strongest reduction in risk of mortality was observed for those with high adherence to the MADP, with an HR of 0.54 (95% CI: 0.37–0.80). The moderate adherence group (HR = 0.65, 95% CI: 0.44–0.96) and the abstention group (HR = 0.60, 95% CI: 0.36–0.98) also exhibited lower risks of mortality than the low MADP adherence group. Conclusions: based on the available evidence, a public health message can be provided to people older than 50 years as follows: among those who drink alcohol, high adherence to the MADP score could substantially reduce their risk of all-cause mortality.

## 1. Introduction

The association between alcohol consumption and risk of disease or mortality is a controversial issue for which no solid evidence exists. Alcohol is a widely consumed substance worldwide [1,2], especially in Europe [3,4]. A huge body of observational prospective epidemiologic studies have consistently suggested a J-shaped association of alcohol with cardiovascular disease or all-cause mortality [5,6,7,8,9]. Light or even moderate consumption has been associated with cardiovascular health benefits such as an increase in HDL cholesterol, reduced platelet aggregation, and antifibrillatory effects [10,11,12,13]. Other studies have also reported inverse associations between moderate consumption and morbidity and even cancer [14,15,16]. However, despite all the prospective observational studies that have been carried out, the underlying mechanisms are not entirely clear, and a strong dose–response relationship may exist [17,18,19]. Harmful alcohol use is, among other things, linked to traffic accidents, intimate-partner violence, suicide, mental illness, liver disease, and cancers of mouth, pharynx, larynx, esophagus, liver, colon, rectum, and breast [1]. With the available evidence, some health agencies and organizations advocate for zero alcohol consumption, especially for young people [18,20,21,22,23].

The study of alcohol consumption is more extensive than simply evaluating the amount of alcohol consumed, and it involves different aspects such as the type of drink or the pattern of consumption [24,25,26,27,28]. Interestingly, an individual’s drinking pattern may act as a strong effect modifier. For this reason, just as the study of dietary pattern versus nutrients separately is becoming more important, alcohol consumption should be assessed with the same methodology [29]. In the Mediterranean diet, there is a pattern of alcohol consumption characterized by low-to-moderate amounts of wine with meals [30]. Furthermore, alcohol is a component of the pattern that contributed the most to the association between adherence to the Mediterranean diet and reduced mortality (24%) [31]. However, despite these data on the importance of the Mediterranean pattern of alcohol consumption on mortality, few studies have assessed the effect of this pattern as a whole [32,33,34,35]. Most of the available evidence studied each component of the Mediterranean pattern separately: low–moderate consumption [36,37,38], preferably at meals [39,40], in the form of wine [41,42,43,44], and spread out over the whole week, avoiding binge drinking [45,46,47,48,49].

In order to study the Mediterranean pattern of alcohol consumption in more detail, as well as its association with mortality, the methodology of a priori dietary patterns, previously defined by the Spanish cohort “Seguimiento Universidad de Navarra” (SUN) was used. The MADP score defined by Gea, A. et al. [33] in the SUN cohort was used to review the association of this pattern of consumption with mortality in the over-50 age group, where there is greater controversy in the recommendations [50]. This score evaluated the Mediterranean patterns of drinking alcohol with seven dimensions: moderate alcohol consumption, preferably wine, during meals, opting for red wine over other wines, low spirits consumption, and alcohol consumption spread out over one week, avoiding binge drinking [33]. Our hypothesis was that high adherence to the MADP may reduce total mortality in men older than 50 years and women older than 55 years. With our results, we hoped to address a more comprehensive aspect of alcohol consumption and its benefits on total mortality in this age group where recommendations on alcohol are controversial.

## 2. Materials and Methods

### 2.1. Study Population

The SUN (“Seguimiento Universidad de Navarra”) study is a dynamic, prospective, multipurpose cohort of highly educated participants (university graduates). Participants report information about their lifestyle, diet, and diseases in a baseline questionnaire, which is updated biennially. Figure 1 shows the selection of the analytical sample. Up to May 2022, 23,133 subjects had completed the baseline questionnaire. For the present analysis, 19,762 women younger than 55 years old or men younger than 50 years old at baseline were excluded. Likewise, 30 participants with insufficient follow-up time, 241 subjects with total energy intake out of predefined limits (<800 or >4000 Kcal/day among men and <500 or >3500 Kcal/day among women) [51], and 774 participants with prevalent diseases (cardiovascular disease, depression, and cancer) were excluded. Among the remaining 2326 subjects, 2226 were successfully followed-up on (overall retention: 95.7%).

This study was approved by the Institutional Review Board of the University of Navarra. Details of the design and methods of this cohort study have been described elsewhere [52].

### 2.2. Mediterranean Alcohol-Drinking Pattern (MADP)

Alcohol consumption at baseline was obtained through a validated 136-item semiquantitative Food Frequency Questionnaire (FFQ) [53]. Information on drinking habits during the year preceding the completion of the baseline questionnaire was also collected.

With this information, the MADP score [33] was used to assess adherence to the MADP. This score has 0 to 9 points in which adherence to each of the following items is assessed: (1) moderate total alcohol intake (alcohol consumption of 5–25 g/day in women and 10–50 g/day in men) is positively scored with 2 points; intakes below this range (>0–5 g/day in women and >0–10 g/day in men) are assigned 1 point, and intakes above this range (>25 g/day in women and >50 g/day in men) are assigned 0 points, (2) preferring wine (at least 75% of alcohol consumed as wine) is scored with 1 point, (3) selecting red wine over other types of wine (at least 75% of wine consumed as red wine) is scored with 1 point, (4) consuming wine preferentially during meals (at least 75% of wine consumed during meals) is positively scored with 1 point, (5) low spirits consumption (lower than 25% of total alcohol intake) is scored with 1 point, (6) alcohol intake spread out over one week (ratio between number of drinking days per week and total g/week of alcohol intake categorized in quartiles) is scored with 2 points for participants in the highest quartile, 1 point for participants in the third and second, and 0 points for those in the lowest quartile, and (7) avoidance of excess drinking occasions (maximum number of drinks consumed on a single occasion never exceeds five drinks) is positively scored with 1 point [33]. These cut-off points were selected considering the previous publications on the Mediterranean drinking patterns [30,33]. The MADP score was grouped into three categories: 0–3 points (low adherence), 4–5 points (moderate), and 6–9 (high adherence). Abstainers, who reported not drinking alcohol, were excluded from this MADP, and they were classified in a fourth group.

### 2.3. Outcome Assessment

The primary outcome was all-cause mortality. Continuous contact with participants was maintained in order to detect each death. Reports from the closest relative, work associates, and the postal system allowed the identification of more than 85% of the deaths. For the rest of the deaths, we reviewed the National Death Registry at least once each year to confirm the vital status and the causes of death of all participants.

### 2.4. Covariate Assessment

We gathered information about sociodemographic and anthropometric variables such as age, sex, marital status, height, and weight [54]. The baseline questionnaire also collected medically diagnosed conditions and lifestyle information including physical activity [55], smoking habit, and alcohol consumption variables. Finally, the validated 136-item FFQ included at baseline [56] was used to compute adherence to Mediterranean diet using the Mediterranean Diet Score (MDS) [57], but we excluded alcohol intake to avoid overlap with our main exposure.

### 2.5. Statistical Analysis

Cox proportional hazard models were used to estimate the association between each category of the MADP score adherence and total mortality. The proportional hazards assumption was met as assessed by the Schoenfeld residuals. Multivariable-adjusted hazard ratios (HR) and 95% confidence intervals were calculated using the low-adherence group as the reference category. For these analyses, we excluded men who were younger than 50 years old and women younger than 55 years old at baseline. In the Cox regression models, the enter time was the date of the baseline questionnaire reception and the exit time was the date of death (for deceased participants) or last contact (for survivors). Age was the underlying time variable (birthday as origin). Multivariate models were stratified by age groups (10-year periods) and year of recruitment (1999/2003, 2004/2009, and ≥2010). Models were also adjusted for sex, baseline body mass index (BMI, kg/m^2^; continuous) using a linear and quadratic term, adherence to Mediterranean Diet Score (3 categories), physical activity (MET-h/week; continuous), marital status (3 categories), number of metabolic conditions at baseline (hypercholesterolemia, hypertriglyceridemia, diabetes, obesity, and hypertension; 4 categories), smoking (combination of categories of smoking habit (never, current, and former), and total cumulative exposure to cigarette smoking (pack-years; 4 categories).

We also evaluated the association of a continuous 2-point increment in the MADP score with the risk of mortality in the Cox regression models after removing the abstainers. Furthermore, tests for linear trend were also performed.

Finally, we conducted sensitivity analyses by evaluating the models under different assumptions.

All *p* values were two-tailed. The level of confidence was 95 for confidence intervals (95% CI). The analyses were performed using STATA/SE version 15.1 (Stata Corp, College Station, TX, USA).

## 3. Results

The baseline characteristics of the participants are presented in Table 1 according to their MADP adherence group. Among the drinkers, participants with high adherence to the MADP drank on average fewer grams of alcohol per day (12.9 (SD 9.9)), they were less likely to be current smokers, they were older (58.4 (SD 6.3)), and most of them were married (82.1%). Moreover, they had the highest physical activity expressed in MET-h/week, even compared to abstainers.

During a median follow-up of 14 years (interquartile range: 9.00–17.13), 291 participants (88.6% men) died. The leading cause of death was cancer (43.3%), followed by other causes (32.3%) and CVD (22.3%). The mean age at death was 75.94 (SD 10.20) years.

The mortality hazard ratios (HR) according to MADP score are shown in Table 2. High adherence to MADP score (6–9 points) showed the lowest risk of death after adjusting for confounders (HR = 0.54, 95% CI (0.37–0.80)) when compared with low MADP adherence. Likewise, moderate adherence to the MADP (HR = 0.65, 95% CI (0.44–0.96)) and abstention (HR = 0.60, 95% CI (0.36–0.98)) presented a reduction in the risk of mortality.

In addition, when we studied the association between MADP as a continuous variable and the risk of mortality, we found that a two-point increment in the score was inversely associated with the risk of mortality, but it lost its significance in the multiple-adjusted model. Moreover, when we conducted trend tests for the MADP among drinkers, we observed a significant inverse linear trend (*p* = 0.003).

In Table 3, we show the results from the sensitivity analyses. Stronger reductions in the risk of mortality were observed for the following scenarios: when we excluded deaths in the two first years (HR = 0.52, 95% CI (0.35–0.78)) and when we only included cancer deaths (HR = 0.46, 95% CI (0.26–0.81)).

## 4. Discussion

As previously described by Gea, A et al., better adherence to the Mediterranean alcohol drinking pattern (MADP) was associated with a lower risk of all-cause mortality [33]. There are several studies studying different aspects of the MADP separately and their association with mortality, but the available evidence for this overall pattern is very scarce [26,28,33,34,35,39]. The inverse association found is consistent with previous studies that evaluated different aspects of the MADP. Mediterranean alcohol consumption is characterized by low–moderate intakes of wine with meals [58]. The distribution of mortality risk and alcohol consumption has been represented in the literature as having a J-shape [5]. This distribution attributes to low–moderate consumption a lower risk of all-cause mortality [5,6,7,8], even in the elderly population [9]. Likewise, moderate consumption has been inversely associated with the incidence of diabetes and CVD [36,59]. Another point in the pattern is the consumption of wine, preferably red wine, as opposed to other beverages. The high presence of polyphenols in wine, especially in red wine, is responsible for its antioxidant and anti-inflammatory capacity. This antioxidant component seems to explain its protective effect against mortality, CVD, and cancer [41,42,43,44,60,61,62,63,64]. In addition, wine intake with meals adds another aspect to the pattern, with studies finding decreased risks of mortality [39], aero-digestive cancers [60], diabetes incidence [40], and cirrhosis [25]. They also report a “washing effect” when alcohol is consumed with food due to the shorter duration in the digestive tract arising from chewing and swallowing [60]. Finally, there is an additional point in the MADP, which is to spread alcohol consumption over the week, avoiding occasional high intakes. This way of distributing alcohol intake is associated with a lower risk of mortality [47,48,49,65]. Furthermore, this inverse association makes biological sense as it avoids the large amounts of alcohol in the blood that trigger oxidative and cardiotoxic mechanisms [10,11]. With these findings, the study of alcohol intake in a patterned way can better summarize the actual consumption as it considers possible synergies and interactions. Moreover, an alcohol drinking pattern can be used to make better public health recommendations [29,66].

Thus far, we have seen that the pattern of alcohol consumption has a differential influence on occurrences of disease and mortality. However, there are other aspects such as age that modify this association. Currently available evidence recommends abstention from alcohol for young people [1,67,68] because when they drink, they tend to do so in unhealthy patterns, such as binge drinking [69]. The 50-year or older age group, however, is more controversial, and there is no strong evidence in favor of complete abstention or moderation [9,22,67,70,71]. Until there are clinical trials that may provide solid evidence, observational approaches will be used in clinical practice [50]. Our study, therefore, aimed to provide evidence for this controversy. For this reason, we limited our analysis to men over 50 years old and women over 55 years old. In this age group, which generates such disparate opinions, our study found that those with high adherence to the MADP significantly reduced their risk of mortality by 46% (HR = 0.54, 95% CI (0.37–0.80)). We also found a dose–response effect in terms of adherence to the MADP (moderate HR = 0.65, 95% CI (0.44–0.96)); high adherence HR = 0.54, 95% CI (0.37–0.80)), with a significant inverse linear trend (*p* = 0.003). Therefore, among drinkers, those with moderate adherence, and especially those with high adherence, have a lower risk of mortality than those with low adherence. This may be because low adherence to the MADP implies a more sporadic, heavy, and distant consumption of wine at meals. These results are consistent with previous studies on this score [32,33,34,35]. On the other hand, in relation to the abstainers’ group, a lower protective effect is observed compared to the low adherence category (HR = 0.60, 95% CI (0.36–0.98)). In addition, regarding the reference group chosen, it was not considered appropriate to choose the group of abstainers for comparison as there could be some bias due to former drinkers and avoidance of alcohol because of previous medical conditions (“sick-quitter” hypothesis) [72]. Therefore, using the low MADP score group as reference, we can say that among those who drink, the risk of all-cause mortality decreases for those who have a higher adherence to the MADP. If we compare the low MADP score to abstainers, we can see that not drinking alcohol is more protective than inadequate drinking (HR = 0.60, 95% CI (0.36–0.98)). Finally, another aspect regarding adherence to this pattern is the decrease in the risk of mortality for each additional two points in MADP score observed in the age-adjusted (HR = 0.84, 95% CI (0.74–0.94)) and age- and sex-adjusted (HR = 0.87, 95% CI (0.77–0.98)) models. However, this protective effect was lost when adjusting for multiple confounders (HR = 0.90, 95% CI (0.79–1.02)). These results may indicate a non-linear relationship or the need for more statistical power.

Although we acknowledge a possible interaction between the MADP score (three categories) and the total amount of alcohol consumed (g/d), our sample is likely to be underpowered for assessing the interaction. We excluded abstainers in this analysis, and we did not find any interaction between MADP and grams of alcohol, with *p* = 0.22.

After various sensitivity analyses, a majority of the results provided similar values to the main analysis. The assumptions where deaths in the first 2 years of follow-up were eliminated (HR = 0.52, 95% CI (0.35–0.78)) and when only cancer deaths were considered (HR = 0.46, 95% CI (0.26–0.81)) indicate a higher magnitude of the protective effect. In addition, when only women and never-smokers were analyzed, a greater protective effect was observed than in the main analysis, but this time it was not statistically significant. In the analysis with only cancer deaths, a statistically significant 62% reduction in the risk of mortality was observed. These results may seem contradictory to the available evidence on the harmful effects of alcohol on cancer [1,4,73,74,75]. However, these studies consider only the grams of alcohol consumed, not the pattern as studied in this analysis. It should also be noted that this result arises from comparison with the category of low adherence to the Mediterranean alcohol drinking pattern, reflecting that moderate wine consumption with meals that is spread out over one week reduces the risk of cancer mortality compared to both higher and occasional consumption. Another possible explanation could be that participants with greater adherence to the MADP also adhere better to healthier lifestyles and, therefore, have a lower risk of cancer mortality. Finally, although it did not yield statistically significant results, the analysis of never-smokers reflected a 62% risk reduction, which is consistent with the available evidence on alcohol and tobacco interaction [76,77,78].

Our study had some limitations that should be mentioned. The SUN cohort is not representative of the general population; therefore, its external validity could be limited. Nevertheless, generalizability of the results should be based on biological plausibility. Second, our cohort is characterized as a young cohort, and so we have less participants with more than 50 years at baseline. However, although this reduction in the sample size reduces the statistical power, statistically significant results have been found. Third, study variables were self-reported, and so some degree of misclassification is possible. Alcohol consumption could be affected by this misclassification; however, in previous validation sub-studies for the dietary habits questionnaire [53], the correlation coefficient of alcohol was high (r = 0.88). Furthermore, we previously evaluated alcohol misclassification and we found that it was not differential [67]. Fourth, the SUN cohort is a university graduate cohort, and so this could affect some aspects of diet and lifestyle. Nevertheless, if a protective effect has been found in this population with healthier lifestyles, in the general population, the protective effect of the MADP could be even greater. Fifth, although the cut-off points established for this study (50 and 55 years) are based on the evidence available from other cohort studies and meta-analyses [70,79,80], our selected age range might not be the ideal setting. Lastly, residual confounding cannot be excluded.

The strengths of this study reside in the long follow-up time, the high retention proportion, and the adjustment for a wide number of potential sources of confounding. Furthermore, being a university graduate cohort could prevent confounding by education and some social factors. On the other hand, as it is a highly educated and motivated cohort, a higher degree of self-reported data quality is assumed.

## 5. Conclusions

This article reflects an important and applicable public health message: if you drink alcohol, follow the Mediterranean pattern. Moderate red wine consumption at meals which is spread throughout the week, avoiding binge drinking, reduces the risk of all-cause mortality by 48%. These results are consistent with individual studies of each separate aspect of the pattern and with studies of a priori patterns. Therefore, the pattern approach to the study of alcohol consumption should be more widely used for analyses than simply grams of alcohol consumed. Thus, it should be recommended that if alcohol is consumed, it should be done in accordance with the MADP.

## Figures and Tables

**Figure 1 nutrients-14-05310-f001:**
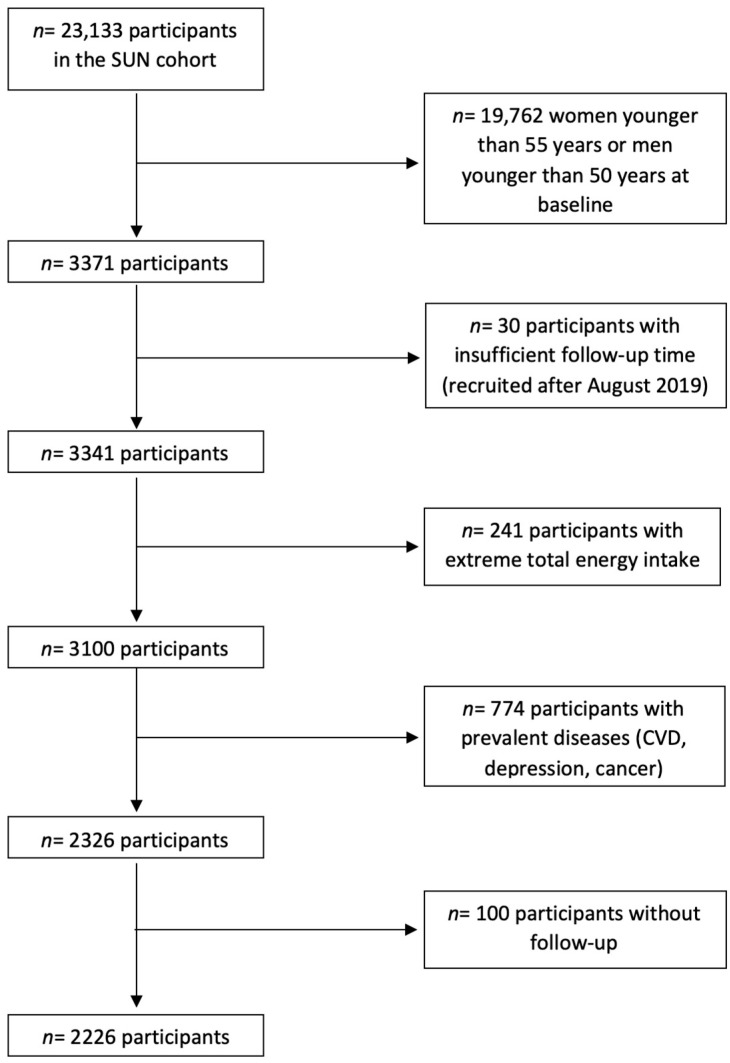
Flow chart of the participants in the “Seguimiento Universidad de Navarra” (SUN) project.

**Table 1 nutrients-14-05310-t001:** Baseline characteristics of participants.

	Mediterranean Alcohol-Drinking Pattern			
	Abstainers	Low (0–3)	Moderate (4–5)	High (6–9)
*n*	250	170	789	1017
Alcohol intake (g/day)	0	35.8 (31.7)	13.9 (13.6)	12.9 (9.9)
Men (%)	56.0	93.5	87.6	80.8
Age (years) (mean (SD))	60.2 (7.7)	57.8 (6.7)	58.1 (6.7)	58.4 (6.3)
BMI (kg/m^2^) (mean (SD))	25.5 (3.9)	26.7 (3.3)	26.5 (3.3)	25.9 (3.3)
Smoking habit (packs/year)				
Non-smokers (%)	46.4	16.4	28.8	32.5
Former smokers (%)	41.6	60.6	52.2	50.7
Current smokers and <10 packs/year (%)	0.8	5.3	3.6	2.9
Current smokers and ≥10 packs/year (%)	11.2	17.7	15.4	13.9
Mediterranean diet score (Trichopoulou, 0 to 8)				
<3 points	26.4	28.8	22.2	19.0
3–5 points	50.8	55.9	58.9	60.7
>6 points	22.8	15.3	18.9	20.3
Number of metabolic conditions ^1^				
No metabolic conditions (%)	38.8	34.1	36.0	38.6
One metabolic condition (%)	34.8	30.6	32.3	32.7
Two metabolic conditions (%)	18.4	20.0	22.3	19.0
Three or more metabolic conditions (%)	8.0	15.3	9.4	9.7
Physical activity (MET-h/week) (mean (SD))	21.6 (27.5)	20.7 (21.4)	22.5 (23.9)	24.3 (21.9)
Marital status				
Single (%)	19.6	8.2	8.6	9.9
Married (%)	70.0	84.1	82.2	82.1
Other (%)	10.4	7.7	9.2	8.0

^1^ Metabolic conditions at baseline: hypercholesterolemia, hypertriglyceridemia, diabetes, obesity, and hypertension.

**Table 2 nutrients-14-05310-t002:** Mortality hazard ratios (HR) and their confidence intervals (95% CI) according to the categories of the MADP score and for each two-point increment.

	Mediterranean Alcohol-Drinking Pattern				Only Among Drinkers	
	Abstention	Low (0–3)	Moderate (4–5)	High (6–9)	Continuous (+2 Points in MADP)	*p* for Trend
Cases/person-years	35/3122	35/2207	102/10,113	119/13,316		
Age-adjusted	0.46 (0.28–0.74)	1 (ref.)	0.61 (0.41–0.90)	0.50 (0.34–0.74)	0.84 (0.74–0.94)	0.001
*p*	0.001		0.013	0.001	0.004	
Age- and sex-adjusted model	0.53 (0.33–0.87)	1 (ref.)	0.63 (0.43–0.94)	0.53 (0.36–0.78)	0.87 (0.77–0.98)	0.002
*p*	0.012		0.022	0.001	0.026	
Multivariable-adjusted model *	0.60 (0.36–0.98)	1 (ref.)	0.65 (0.44–0.96)	0.54 (0.37–0.80)	0.90 (0.79–1.02)	0.003
*p*	0.042		0.030	0.002	0.093	

* Adjusted for sex, BMI (kg/m^2^), physical activity (MET-h/week), smoking habit (four groups and packages/year, four categories), punctuation in the Mediterranean Diet Score at baseline (three categories), number of metabolic conditions at baseline (hypercholesterolemia, hypertriglyceridemia, diabetes, obesity, and hypertension; three categories), and marital status (five categories).

**Table 3 nutrients-14-05310-t003:** Sensitivity analyses: Association of high adherence to MADP score and total mortality under a diversity of scenarios.

	Cases/Person-Years	High (6–9) vs. Low (0–3)
Main analysis	291/28,759	0.54 (0.37–0.80)
Only men	258/23,659	0.57 (0.38–0.86)
Only women	33/5099	0.22 (0.02–2.87)
Including alcohol in the MDS *	291/28,759	0.56 (0.37–0.83)
Excluding those with prevalent metabolic conditions (hypercholesterolemia, hypertriglyceridemia, diabetes, and hypertension)	86/11,691	0.75 (0.36–1.54)
Only never-smokers	68/9344	0.37 (0.11–1.25)
Excluding deaths in two first years	272/24,328	0.52 (0.35–0.78)
Only cancer deaths	126/26,761	0.46 (0.26–0.81)
Only CVD deaths	65/26,145	0.69 (0.29–1.65)
Only other causes of death	100/26,705	0.57 (0.26–1.23)

* MDS: Mediterranean diet score (9 points).

## Data Availability

The data presented in this study are available on request from the corresponding author.

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
