# Peer review of "Mediterranean Alcohol-Drinking Patterns and All-Cause Mortality in Women More Than 55 Years Old and Men More Than 50 Years Old in the “Seguimiento Universidad de Navarra” (SUN) Cohort"

_nutrients, 2022, doi:10.3390/nu14245310_

Round 1
Reviewer 1 Report
The study reported in this submission investigate the effect of a mediterranean alcohol-drinking pattern (MADP) in a cohort of 2,226 middle age, male and female individuals from Spain on all-cause mortality risk. Participants to the study were divided inot low, moderate, and high categories based on their adherence to MADP criteria, and compared to abstainers. The results reported in the study indicate the the strongest reduction in mortality risk was observed in individuals with the higher adherence of the MADP criteria. The moderate category and the abstainers also showed a decreased mortality risk as compared to individuals in the low adherence category. This conlusion is supported by the data presented in the study.
Author Response
The study reported in this submission investigate the effect of a mediterranean alcohol-drinking pattern (MADP) in a cohort of 2,226 middle age, male and female individuals from Spain on all-cause mortality risk. Participants to the study were divided inot low, moderate, and high categories based on their adherence to MADP criteria, and compared to abstainers. The results reported in the study indicate the the strongest reduction in mortality risk was observed in individuals with the higher adherence of the MADP criteria. The moderate category and the abstainers also showed a decreased mortality risk as compared to individuals in the low adherence category. This conlusion is supported by the data presented in the study.
Thank you very much for your time in reviewing our manuscript. We hope that the results presented have been valuable and interesting to you.
Reviewer 2 Report
The authors studied the associations between Mediterranean alcohol-drinking pattern and all-cause mortality in participants with older ages. They concluded that high adherence to the Mediterranean alcohol-drinking pattern score could substantially reduce the risk of all-cause mortality, which may provide valuable information for alcohol consumers in this populations. But there are a few points or needs clarification.
In the abstract, Line 20-21, This sentence may lead to confusion. In fact, participants were divided into 4 groups in this study. More clear expression could be used here.
Irrelevant keywords should be removed, such as cancer and cardiovascular disease.
The classification basis of MADP score may need to be explained.
This study just included women older than 55 years old and men older than 50 years old. How did the researchers determine the age cutoff points? Authors may need to discuss potential bias on this.
In discussion, line 224-225, “There is also a dose-response effect in terms of adherence to the pattern”. A trend test may need here.
In Table 1 and 2, a top line needs to be added.
Test for Proportional hazard assumption of Cox regression need to be described in the paper.
Author Response
The authors studied the associations between Mediterranean alcohol-drinking pattern and all-cause mortality in participants with older ages. They concluded that high adherence to the Mediterranean alcohol-drinking pattern score could substantially reduce the risk of all-cause mortality, which may provide valuable information for alcohol consumers in this populations. But there are a few points or needs clarification.
In the abstract, Line 20-21, This sentence may lead to confusion. In fact, participants were divided into 4 groups in this study. More clear expression could be used here.
Thank you for your suggestion. We agree and we have rephrased the sentence.
“We classified participants in 3 categories of adherence to the MADP score (low, moderate, high) and we added a fourth category for abstainers.”
Irrelevant keywords should be removed, such as cancer and cardiovascular disease.
Thank you for your comment. We have removed irrelevant keywords (cancer and cardiovascular disease)
The classification basis of MADP score may need to be explained.
In line with your suggestion, we complete the explanation of the MADP score.
“This score has 0 to 9 points in which adherence to each of the following items is assessed: (1) moderate total alcohol intake (alcohol consumption of 5-25g/day in women and 10-50 g/day in men is positively scored with 2 points; intakes below this range (>0-5g/day in women and >0-10g/day in men) are assigned 1 point and intakes above this range (>25g/day in women and >50g/day in men) are assigned 0 points), (2) preferring wine (at least 75% of alcohol consumed as wine) is scored with 1 point, (3) selecting red wine over other types of wine (at least 75% of wine consumed as red wine) is scored with 1 point, (4) consuming wine preferentially during meals (at least 75% of wine consumed during meals) is positively scored with 1 point, (5) low spirits consumption (lower than 25% of total alcohol intake) is scored with 1 point, (6) alcohol intake spread out over the week (ratio between number of drinking days per week and total g/week of alcohol intake categorized in quartiles; 2 points for participants in the highest quartile, 1 point for participants in the third and second and 0 points for the lowest quartile), and (7) avoidance of excess drinking occasions (maximum number of drinks consumed on a single occasion never exceeded five drinks) is positively scored with 1 point [33]. These cut-off points were selected considering previous publications on the Mediterranean drinking pattern [30,33]. The MADP score was categorized into three categories: 0-3 points (low adherence), 4-5 points (moderate), and 6-9 (high adherence). Abstainers, who reported not drinking alcohol, were excluded from this MADP and they were classified in a fourth group.”
This study just included women older than 55 years old and men older than 50 years old. How did the researchers determine the age cutoff points? Authors may need to discuss potential bias on this.
Thank you for your comment. We have added a limitation in the discussion section about this topic.
Fifth, although the cut-off points established for this study (50 and 55 years) are based on the evidence available from other cohort studies and meta-analyses [70, 79, 80] there may be a potential bias.
In discussion, line 224-225, “There is also a dose-response effect in terms of adherence to the pattern”. A trend test may need here.
According with your suggestion, we included trend tests in our analyses.
“There is also a dose-response effect in terms of adherence to the pattern (moderate HR=0.65, 95% CI (0.44-0.96)); high adherence HR=0.54, 95% CI (0.37-0.80)), with a significant inverse linear trend (P=0.003).”
In Table 1 and 2, a top line needs to be added.
Thank you for pointing this out. We added both lines in the tables.
Test for Proportional hazard assumption of Cox regression need to be described in the paper.
In line with your suggestion, we have added a sentence describing this in the Methods section.
"Proportional hazards assumption was checked with Schoenfeld residuals”